# Integrated Evaluation of the Aeroacoustics and Psychoacoustics of a Single Propeller

**DOI:** 10.3390/ijerph20031955

**Published:** 2023-01-20

**Authors:** Jianwei Sun, Koichi Yonezawa, Eiji Shima, Hao Liu

**Affiliations:** 1Graduate School of Engineering, Chiba University, Chiba 263-8522, Japan; 2Central Research Institute of Electrical Power Industry, Abiko 270-1194, Japan; 3Center for Aerial Intelligent Vehicles, Chiba University, Chiba 263-8522, Japan; 4Japan Aerospace Exploration Agency, Tokyo 181-0015, Japan

**Keywords:** loop-type propeller, aerodynamic noise, psychoacoustic, broadband noise, tonal noise

## Abstract

Aeroacoustic noise in multiple rotor drones has been increasingly recognized as a crucial issue, while noise reduction is normally associated with a trade-off between aerodynamic performance and sound suppression as well as sound quality improvement. Here, we propose an integrated methodology to evaluate both aeroacoustics and psychoacoustics of a single propeller. For a loop-type propeller, an experimental investigation was conducted in association with its aerodynamic and acoustic characteristics via a hover stand test in an anechoic chamber; the psychoacoustic performance was then examined with psychoacoustic annoyance models to evaluate five psychoacoustic metrics comprising loudness, fluctuation strength, roughness, sharpness, and tonality. A comparison of the figure of merit (FM), the overall sound pressure level (OASPL) and psychoacoustic metrics was undertaken among a two-blade propeller, a four-blade propeller, the loop-type propeller, a wide chord loop-type propeller, and a DJI Phantom III propeller, indicating that the loop-type propeller enables a remarkable reduction in OASPL and a noticeable improvement in sound quality while achieving comparable aerodynamic performance. Furthermore, the psychoacoustic analysis demonstrates that the loop-type propeller can improve the psychological response to various noises in terms of the higher-level broadband and lower-level tonal noise components. It is thus verified that the integrated evaluation methodology of aeroacoustics and psychoacoustics can be a useful tool in the design of low-noise propellers in association with multirotor drones.

## 1. Introduction

In recent years, UAVs (unmanned aerial vehicles), particularly multi-rotorcrafts, namely, drones have been broadly recognized as playing a crucial role in impacting civilian tasks, including urban logistics, airborne imagery and even transportation [1]. They have increasingly attracted much interest in civil applications [2,3] and academic research, where an important focus of such research is the propulsion system and approaches of optimizing their aerodynamic and aeroacoustic performance [4,5,6]. UAVs have emerged as useful platforms for flying over or hovering in cities where urban populations live. However, given the rapidly increasing needs and applications of various drones, a crucial issue that has emerged is the rotor noise, which seriously disturbs the environment and community [7,8,9].

As the major sound source of rotorcrafts, the noise emission characteristics of propulsion systems, practically propellers, have been the main subject to be explored in many previous studies [10,11,12,13,14,15,16,17,18], for the purpose of reducing the propellers’ aerodynamic noise, comprising tonal noise and broadband noise [19]. The broadband noise generally originates from three different sources: the noise caused by the formation and shedding of vortices around blades, the noise produced directly by the turbulent boundary layer interacting with the trailing edge of the blade, and the noise induced by the flow separation on the airfoil surface. The tonal noise results from the unsteadiness of the periodical pressure fluctuations due to the rotational blade [4,19]. Thus, it is important to investigate the various components in association with the tonal noise and broadband noise so as to quantify the noise characteristics of drone propellers. To suppress the noise generation in the overall sound pressure level (OASPL), which is broadly used to evaluate the intensity of noise samples, efforts on propeller design are usually made in applying bioinspired serration structures [10,11,12], add-in structures to control fluid fields [13,14,18], and advanced control methods [15,16]. Recently, we proposed a low-noise loop-type propeller that can generate lift forces with loops rather than conventional blades and achieved a significant reduction in the OASPL [17]. The OASPL is usually calculated through power spectral density integration on the human auditory band (usually 20–20,000 Hz) in the frequency domain. It, however, cannot fully reflect the actual impact of the drones’ noise emission on human beings because the noise response and feeling of humans also greatly depends on the acoustic characteristics in the frequency domain [20].

In terms of psychoacoustics, Merino-Martinez et al. (2021) [21] presented a psychoacoustic metrics-based approach to evaluate the wind turbine acoustic performance of noise suppression measures, and pointed out the importance of the noise characteristics for the perceived annoyance, including the tonality, spectral content, and amplitude modulation. In fact, a series of studies have shown that the noise generated by rotorcrafts can cause human beings to be more annoyed than the traffic noises induced by other transportation sources because of the very specific noise characteristics of the drone propellers [9,22,23,24,25,26]. In the psychoacoustic study of the noise emission characteristics of drone propellers, i.e., the rotor noise, both objective and subjective evaluations are combined to quantify the psychological response to the drone noise. Christian and Cabell [9] defined the psychoacoustic sound exposure level offset with the A-weight OASPL difference of noise samples for the same annoyance level, and used it in evaluating the annoyance in response to the noises of road vehicles and drones. Gwak et al. [24] proposed a sound-quality factor to quantify the annoyance of hovering drones based on two psychoacoustic subjective experiments and found that the loudness, the sharpness, and the fluctuation strength are significantly high in the case of UAV noise. Schäffer et al. [27] gave a systematic review on both the noise emission and its impact on human beings. Torija et al. [26,28] investigated the aerodynamic interaction effects on the noise production by UAVs with contrarotating propellers by means of psychoacoustic analysis and further developed an optimized psychoacoustic annoyance model for the rotor-induced noises based on the subjective listening experiment. Furthermore, Torija et al. [8] gave an extensive discussion on the potential and feasibility to develop a psychoacoustics-informed system to greatly constrain the harmful impact of UAV noises on human communities. The psychoacoustic performance, however, remains poorly studied, particularly in association with the quantitative evaluation of the aeroacoustic characteristics of drone propellers in terms of psychoacoustic metrics, and its applications in low-noise propeller design with consideration of both aerodynamic performance and human psychological response.

In this paper, we aim at proposing an integrated methodology to evaluate both aero-acoustics and psychoacoustics of a single propeller, which is utilized to experimentally investigate the aerodynamic and aeroacoustic characteristics of five different prototype propellers while analyzing their psychoacoustic performance by examining psychoacoustic annoyance using a model based on five psychoacoustic metrics comprising loudness, fluctuation strength, roughness, sharpness, and tonality. The remainder of this paper is organized as follows. Section 2 introduces the experimental setup and methodology. Section 3 and Section 4 compares and discusses the aerodynamic performance results, the aeroacoustics, and the psychoacoustic performance results, where the effective improvement of the loop-type propeller in terms of psychoacoustic annoyance is presented. Section 5 summarizes the findings in this study.

## 2. Methods

### 2.1. The Hover Stand Setup for Aeroacoustic Measurements

Measurements of aerodynamic forces (thrust) and torque and acoustic characteristics were conducted simultaneously using a hover stand setup equipped in an anechoic chamber with a cut-off frequency of 200 Hz, and internal dimensions of 3.86 m (length) × 1.56 m (width) × 2.4 m (height), at Nishi Chiba Campus, Chiba University.

A high-resolution, six degrees-of-freedom (6DoF), force–torque cell (Leptrino PFS 030YA151) was attached between the stand and the motor adapter. A Kyowa EDX-100A data acquisition system with a 10 Hz low-pass digital filter was employed to reduce the noise of the voltage output signal, with a sampling rate ($f_{s}$) of 20 kHz applied. The thrust ($F_{z}$) range of the force–torque cell was ±150 N with a precision of 0.0375 N, corresponding to a torque ($M_{z}$) range of ±1 N·m with a precision of 0.00025 N·m. The hover stand was made from an aluminum frame 800 mm in height. To avoid the downwash-induced ground effect, the propeller was fixed at a negative angle of attack to ensure an upwards induced flow (Figure 1). The microphone array was thus located downstream of the propeller.

All noise signals were collected and recorded with RION CO. UC-59 microphones using a Kyowa EDX-100A data acquisition system at a sampling rate ($f_{s}$) of 20 kHz. The dedicated data acquisition and amplifier channel provided power to the microphones while filtering and digitizing their voltages with a low-pass filter of 10 kHz with a precision of 24 bits. As illustrated in Figure 1, four microphones were arrayed in an arc fashion with a distance of 600 mm (i.e., 2.5d, where d is the diameter of the propellers) from the rotational axis of the propeller. Four polar angles θ are defined as the angles between the stroke plane (i.e., the rotational plane) of the propeller and the line between the microphone and the center of the propeller, which are 0, 25, 50, and 75 deg, corresponding to Mic 1, Mic 2, Mic 3, and Mic 4, respectively.

We further designed a closed-loop controller using an NI USB-6229 and LabVIEW to control the rotational speed or the aerodynamic force, i.e., the thrust generated by the propeller. A tachometer (HIOKI FT3406)/force–torque cell (Leptrino PFS 030YA151) transduced the rotational speed/thrust to the analog signals output, which was used as an input to the closed-loop controller to achieve rotation speed/thrust control. The propulsion system was composed of an EMAX GT2820/06 motor and Hobbywing Platinum 40 A electronic speed controllers (ESCs), driven by an input voltage of 16.8 V (4S lithium battery). At 10 rotational speeds of 3000, 3300, 3600, 3900, 4200, 4500, 4800, 5100, 5400, and 5700 RPM (revolutions per minute), thrust forces, torques, and sound pressures were measured and analyzed. To examine and exclude the background noise caused by the motor noise, we first carried out a pre-experimental test with the motor under all the rotational speeds and quantified the motor-induced noise characteristics. Furthermore, by means of the closed-loop controller, a stable output of the thrusts was confirmed to be capable of producing thrust forces of 2 N, 3 N, 4 N, and 5 N.

The uncertainty associated with the measurements was then examined as an expression of the statistical dispersion of values attributed to a quantity. The overall uncertainty in the thrust measurement was determined by the root sum of the squared uncertainties comprising calibration uncertainty and repeatability uncertainty [6,29,30]. The calibration uncertainty was estimated using the calibration sheet with a series of calibrator-based measurements; the repeatability uncertainty was evaluated in a normal distribution manner based on 10 measurements. The repeatability uncertainty of the thrust and torque measurement was verified to have a 68% confidence level of 0.11 N and 0.0011 N·m, respectively. Using the root sum of the squares of the two sources, 0.12 N and 0.0012 N·m were calculated as the overall uncertainties of the force–torque cell measurement.

The calibration uncertainty of the microphone was measured to be 0.015 dB with a 68% confidence level based on the 10 measurements with the NC-74 calibrator (94 dB, 1 kHz). The propeller rotating measurement with the closed-loop rotation speed experimental setup was conducted up to 10 times, of which the overall sound pressure level (OASPL) uncertainty was estimated to be 0.3 dB at the 1st blade passing frequency (BPF) and 0.61 dB for OASPL with a 68% confidence level.

### 2.2. Propellers

Five propellers, including a two-blade propeller, a four-blade propeller, a wide chord loop-type propeller, a DJI Phantom III propeller, and a loop-type propeller (Figure 2), were employed in the experiments. Note that the loop-type propeller used here is based on a previous study by Shima et al. [17], which was designed with the same diameter as that of the DJI Phantom III propeller (Figure 2e). All the propellers utilized in this study were 3D printed with polyamide (PA) and had the same diameter of 240 mm. The four-blade propeller (#2) (Figure 2b) was used as a reference for a comparison of the blade number effect, which has the same diameter and total blade area as the two-blade propeller (#1) (Figure 2a). The wide chord loop-type propeller (#4) (Figure 2d) has a chord length 10% longer than that of the basic design, i.e., the loop-type propeller (#3). The radial distributions of the normalized chord length and the pitch angle of these five propellers are shown in Figure 3. The chord length at 75% radial position from the center of propellers #1–#5 is 22.8 mm, 11.4 mm, 10.4 mm, 11.5 mm, and 16.2 mm, receptively.

### 2.3. Thrust Coefficient, Torque Coefficient, and Figure of Merit

At a specific rotational speed, all the loads acting on the propeller were measured by a six-force component load cell (Leptrino PFS 030YA151), which was sampled at a frequency of 20 kHz for 10 s using a Kyowa EDX-100A data recorder. The thrust coefficient, $C_{t}$, and the torque coefficient, $C_{Q}$, are nondimensionalized as
(1)$$C_{t\;}=\frac{F_{z}}{\rho {\left(r\Omega \right)}^{2}A},$$
(2)$$C_{Q\;}=\frac{M_{z}}{\rho {\left(r\Omega \right)}^{2}rA},$$
where r is the propeller radius, Ω is the rotation speed in rad/s, ρ is the density of air, A is the disk area of the propeller’s rotational plane, and $F_{z}$ is the measured thrust force. Given the measured torque ($M_{z}$), the figure of merit (FM), which is used to evaluate the propeller’s efficiency, is defined as
(3)$$FM=\frac{C_{T}^{3/2}}{\sqrt{2}C_{Q}}.$$

### 2.4. Sound Signal Data Processing

For the sampling rate ($f_{s}$) of 20 kHz, a lowpass analog filter of 10 kHz was applied in the sound signal sampling through Data AcQuisition (DAQ) to filter the high-frequency signals and to avoid aliasing. The sound signals of a 10 s period were then acquired by the data logger. The spectral makeup of the acoustic waveforms was further characterized by means of power spectral densities (PSDs) based on Welch’s method. The sound signal data of each set had 16,384 (2^14^) samples in length within a rectangular window with 50% overlapping between the records, resulting in a narrow spectral resolution (Δf) of 1.22 Hz. The sound pressure level (SPL) is calculated by Equation (4), where the reference sound pressure ($P_{ref}$) of air is taken as 20 µPa. The overall sound pressure level (OASPL) is defined by Equation (5), where $P_{rms}$ represents the sound pressure root mean square (rms).
(4)$$SPL=10\log_{10}\frac{PSD}{P_{ref}{}^{2}},$$
(5)$$OASPL=20\log_{10}\frac{P_{rms}}{P_{ref}}.$$

The tonal noise and broadband noise component can then be evaluated by the tonal noise $OASPL_{i}$ at the ith-shaft rate frequency ($f_{i},\;i=NB*1,\;2,\;3\ldots$) and the broadband noise $OASPL_{band}$ [13] over a broadband noise range from $f_{L}$ to $f_{H}$, respectively, as follows:(6)$$OASPL_{i\;}=10\log_{10}\frac{\int_{f_{i}-2\Delta f}^{f_{i}+2\Delta f}PSD\;df}{P_{ref}{}^{2}},$$
(7)$$OASPL_{band}=10\log_{10}\frac{\int_{f_{L}}^{f_{H}}PSD\;df}{P_{ref}{}^{2}},$$
where $f_{i}$ is the frequency at the ith harmonic of the shaft rate, Δf is the frequency resolution (1.22 Hz), and PSD is the power spectral density based on Welch’s method.

### 2.5. Objective Psychoacoustic Metrics

Objective psychoacoustic metrics are defined as objective physical quantities to describe the subjective feelings caused by sound [20], normally comprising the four factors of loudness, sharpness, roughness, and fluctuation strength. Here, following the improved Zwicker’s psychoacoustic annoyance model by Di et al. (2016) [31], in which the tonality effect was taken into account, we employed the 5 psychoacoustical parameters of loudness, sharpness, roughness, fluctuation strength, and tonality, and Di’s psychoacoustic annoyance model to evaluate the perceived annoyance of various propeller noise samples.

The loudness (L, unit sone) describes how humans subjectively perceive sound intensity. One sone in loudness is defined as the sound produced by a pure tone of 40 dB at 1 kHz. At 1 kHz, an increase of 10 dB corresponds to a doubling in loudness (sone). Note that this scale of doubling depends on the frequency which is based upon humans’ response. The loudness is a key parameter in objective psychoacoustic assessments, which may be estimated in the manner of the specific loudness of each bark, such as:(8)$$L=\int_{0^{\prime }}^{24Bark}N^{\prime }\left(z\right)dz,$$
where $N^{\prime }$ denotes the specific loudness over a critical-band rate, and Bark ranges from 1 to 24, that is, the frequency band divided by Zwicker according to human response [32].

The sharpness (S, unit acum) is an indicator of the proportion of high-frequency components in a sound. According to Zwicker and Fastl [20], a sound with a sharpness of 1 acum is defined as a narrow band noise one critical band wide with a level of 60 dB at its center frequency of 1 kHz. A signal with more high-frequency components in hearing-spectrum energy will have a greater sharpness value. The sharpness objectively reflects the harshness of the sound signal, which is calculated as
(9)$$S=0.11\frac{\int_{0}^{24Bark}∕N^{\prime }\left(z\right)g\left(z\right)zdz}{\int_{0}^{24Bark}N^{\prime }\left(z\right)dz},$$
where $N^{\prime }\left(z\right)$ is the specific loudness at *z* Bark, and g(z) is the weighting coefficient that depends on *z* which increase by 15.8 Bark (~3 kHz), given by
(10)$$g\left(z\right)=\{\begin{array}{cc} 1, & z\leq 15\\ 0.15e^{0.308\left(z-15\right)}+0.8, & z>15 \end{array}.$$

The roughness (*R*, unit asper) of a sound is quantified by the subjective perception of a rapid amplitude modulation over a frequency range of 15–300 Hz, reflecting the nonstationary degree [20]. In association with the rotational propeller noise, the modulation frequency normally points to the blade passing frequency ($f_{mod}=$ BPF), and thus, the roughness for propeller noise partly reflects the strength at the BPF harmonics and the dynamic balance of the propeller but also affected by impact, structural vibrations, etc. One asper is defined as the roughness produced by a tone (1 kHz, 60 dB) with 100% amplitude modulated at 70 Hz, and the roughness (*R*) is evaluated by
(11)$$R=0.3f_{mod}\int_{0}^{24Bark}f_{mod}\Delta L\;dz,$$
where $f_{mod}$ is the modulation frequency (normally BPF) and ΔL is the perceived modulation depth.

The fluctuation strength (FS, unit vacil) is used to quantify the subjective perception at low modulation frequencies (up to 20 Hz) and is thus similar to the roughness in definition, which describes the fluctuations in slow-moving amplitude modulation of a sound [20]. One vacil is defined as the roughness produced by a tone (1000 Hz, 60 dB) with 100% amplitude modulated at 4 Hz. The FS based on Zwicker’s model is given as
(12)$$FS=\frac{0.008\int_{0}^{24Bark}\Delta L\;dz}{\left(\frac{f_{mod}}{4}\right)+\left(\frac{4}{f_{mod}}\right)\;},$$
where $f_{mod}$ denotes the modulation frequency and ΔL is the perceived modulation depth. It is worth noting that given the fast rotational speeds of the five isolated propellers, the low frequency-based modulation rarely occurs. The FSs calculated here had much lower values and thus could not affect their psychoacoustic annoyance significantly.

The Aures tonality (T, unit tu) is a psychoacoustic parameter that measures the proportion of pure tone components in a sound signal spectrum [33]. 0 tu points to a noise without any discrete tones, and 1 tu presents a 60 dB sine tone at 1 kHz without other noises present. The tonality T is given as:(13)$$T=c\cdot \omega_{T}^{0.29}\cdot \omega_{N}^{0.79},$$
where c is the calibration factor, $\omega_{N}$ is the fraction of the total loudness due to tonal components, $\omega_{T}$ is the extra overall weighting function relevant to the pitch perception.

Since the Zwicker’s psychoacoustic annoyance (PA) model does not include the tonality metric to evaluate the perceptual effects of tonal sound, we herein applied the improved Zwicker’s PA model by Di et al. (2016) [31] to estimate the annoyance degrees of the noise samples. As defined in Equation (14), this model is utilized to evaluate the humans’ subjective feeling of annoyance under various sound conditions.
(14)$$PA=N_{5}\left(1+\sqrt{w_{s}^{2}+w_{FR}^{2}+w_{T}^{2}}\right),$$
$$w_{s}=\{\begin{array}{c} \left(S-1.75\right)\cdot \left(0.25\log_{10}\left(N_{5}+10\right)\right),\;\;S>1.75\\ 0,\;\;S\leq 1.75 \end{array},$$
$$w_{FR}=\frac{2.18}{{\left(N_{5}\right)}^{0.4}}\left(0.4F+0.6R\right),$$
$$w_{T}=\frac{6.41}{{\left(N_{5}\right)}^{0.52}}T.$$
where $N_{5}$ expresses the percentile loudness (sone) with a level exceeding 5% of the time. $w_{s}$ and $w_{FR}$ are the normalized sharpness and fluctuation strength, respectively, with the subscripts of *S*, *R*, *T*, and *FS* denoting the sharpness, roughness, tonality, and fluctuation strength, respectively. This implies that the more annoyed humans feel, the greater the psychoacoustic annoyance is.

## 3. Results

### 3.1. Aerodynamic Performance

Noise suppression and sound quality improvements normally come at the cost of aerodynamic efficiency or thrust production [10,34,35,36]. To verify the loop-type propeller’s trade-off, aerodynamic performance, measurements were conducted simultaneously in the anechoic chamber. Measurements of thrust and torque were acquired with the isolated propeller operating under static rotational speed and static thrust conditions. The thrust coefficient ($C_{t}$), torque coefficient ($C_{Q}$), and figure of merit (FM) of different propellers are shown in Figure 4, where the thrust and torque obtained from the six-degrees-of-freedom force–torque cell are plotted against the rotational speed of these five propellers. The test was performed at ten constant rotational speeds and four constant thrusts. Figure 4a,b plot the thrust and torque coefficients from the two-blade propeller, four-blade propeller, loop-type propeller, wide chord loop-type propeller, and DJI Phantom III propeller, which are shown as #1 to #5, respectively. Our results show a noticeable difference in aerodynamic force production among the target propellers, of which the DJI Phantom III propeller (#5) produced the highest level thrust and torque at the same rotational speed, whereas the loop-type propeller (#3) showed a decrease in both the thrust and torque. In the FM result presented in Figure 4c, the two-blade propellers (#1 and #5) are more efficient in aerodynamic performance than the four-blade propellers (#2, #3, and #4). Note that the FM of the loop-type propeller (#3), which is used to evaluate the hover efficiency, is higher than those of the other two four-blade propellers (#2 and#4).

To further compare the aerodynamic efficiency performance, hover tests on static thrust at stable thrusts of 2 N, 3 N, 4 N, and 5 N were carried out. The mechanical power of the rotational propeller was calculated by the definitions given in Equation (15) [37]:(15)$$P_{\mathrm{mechanical}}=M_{z}\cdot \Omega ,$$
where $P_{\mathrm{mechanical}}$ is the propeller mechanical power in W, Ω is the rotation speed in rad/s, and $M_{z}$ is the measured torque in N·m. Figure 5 compares the mechanical power of the five propellers at four stable thrusts by a closed-loop thrust control system. Obviously, the two-blade propellers (#1 and #5) are more aerodynamically efficient than the four-blade propellers (#2, #3, and #4). It was observed that propeller #1 and propeller #5 produce the same thrust with a lower-level RPM and lower-level mechanical power, which reflects the two-blade propeller rotating with less overall drag.

In summary, the aerodynamic performances of the five propellers are characterized by the thrust coefficient ($C_{t}$) of #5 > #1 > #2 ≈ #4 > #3, the torque coefficient ($C_{Q}$) of #5 > #4 > #2 > #1 > #3, and the figure of merit (FM) of #1 > #5 > #3 > #2 > #4, respectively. The featured aerodynamic performances can be explained by the fact that the profile loss is dominant rather than the induced loss in low-Reynolds number flows. To develop a high-efficiency propeller for small-scale UAVs, one may consider increasing the blade number, which would increase the profile drag, resulting in decreasing the Reynolds number and hence reducing the propeller efficiency [38]. This was in fact observed in the four-blade propellers of #2, #3, and #4. Comparatively, the two-blade propellers (#1 and #5) achieve a high-speed rotation at higher Reynolds numbers, thus showing better aerodynamic performance. Furthermore, the sweep angle at the tip of loop-type propellers (#3 and #4) also affects the thrust production [17].

### 3.2. Aerodynamic Noise Characteristics

#### 3.2.1. Noise Spectra and Overall Sound Pressure Level (OASPL)

To illustrate the propulsion (motor and electric speed controller) acoustic performance more intuitively, in Figure 6, we plotted and compared the noise spectra of the microphone at Mic 2 (θ=25 deg) for a single motor and the DJI Phantom III propeller. As there were 14 magnets in the motor, the 14th shaft rate harmonic (1260 Hz) appears to be the sound source of the motor. Additionally, the noise at a frequency of approximately 3000 Hz is thought to be caused by an electric speed controller (ESC).

Figure 7 shows a comparison of the power spectral density (PSD) of the two-blade propeller (#1), the four-blade propeller (#2), the loop-type propeller (#3), the wide chord loop-type propeller (#4), and the DJI Phantom III propeller (#5) at 5400 RPM, measured at Mic 2 (θ=25 deg). To evaluate the aeroacoustic performance in the frequency domain, each power mean value was calculated over a sliding window with a length of 20Δf (Δf=1.22 Hz), shown in Figure 7b,c. A single-frequency noise of 1260 Hz was produced by the motor. Notably, as shown in Figure 7a, the spectrum indicates that the four-blade propeller (#2) produces more high-frequency broadband noise above 6000 Hz and almost eliminates the tonal noise at $f_{i}=180\;\mathrm{Hz},\;540\;\mathrm{Hz},\;900\;\mathrm{Hz}\ldots$ because of the four-blade propeller. In Figure 7b, the lower-order harmonics of these five propellers are compared at 100 Hz and 4000 Hz. The amplitude of the tone produced by the two-blade propeller (#1) and DJI Phantom III propeller (#5) is obviously larger than that of the other three propellers. Figure 7c shows a close-up view of the PSD of the broadband noise in the high-frequency range, where the shaded areas present the standard deviation of the spectra. The broadband noise SPLs of the loop-type propeller (#3) and the wide chord loop-type propeller (#4) are also shown at the higher levels in the high-frequency range compared to the broadband noise of the two-blade propeller (#1) and the DJI Phantom III propeller (#5). Moreover, it is found that only the two-blade propellers (#1 and #5) produced an obvious higher-order harmonic (i>25∗NB) peak in the PSD spectrum above 4500 Hz (Figure 7c).

Furthermore, the overall sound pressure level (OASPL) values of the five propellers at different locations at 5400 RPM are compared in Figure 8a. All the five propellers present a minimum OASPL at the location of Mic 1 (θ=0 deg), which is likely caused by the dipole sound source directional characteristics [19]. The loop propeller (#3) presents an excellent aeroacoustic feature for the microphone arc array configuration, with the noise OASPL reduced markedly by 0.4–4.7 dB at 5400 RPM compared to the DJI Phantom III propeller (#5), while the wide chord propeller (#4) shows a reduction in the OASPL by 0.6–4.4 dB. It is inferred that the four-blade propeller (#2) has more broadband noise but fewer tonal noise components, whereas the two-blade propeller (#1) has less high-frequency broadband noise but more tonal noise components. Propeller #1 and propeller #2 share a similar OASPL value in the microphone array, with a mere difference between the two propellers of less than 0.5 dB at various locations.

In addition, the same analysis was conducted for the experiment at a stable thrust of $F_{z}=3\;N$ with the changing RPMs. Figure 8b shows the noise OASPLs of the five propellers with the thrust closed-loop control system based on the force–torque cell. The loop-type propeller (#3) obviously presents a markedly low-noise OASPL design capable of achieving a reduction of 1.1–3.5 dB compared to the DJI Phantom III propeller (#5), even though with the loop-type propeller rotating at a higher RPM. Moreover, as expected, the OASPL of the wide chord loop-type propeller (#4) noise also shows a marked improvement in aeroacoustic performance with a pronounced decrease compared to the DJI Phantom III propeller (#5) noise.

#### 3.2.2. Tonal Noise and Broadband Noise

Tonal noise is commonly referred to as discrete frequency noise, which is the BPF (blade-passing frequency) and harmonics frequency noise of the rotational propeller. Broadband noise points to a noise in which sound energy is distributed over a wide section [19]. The effects of the five propellers on the tonal noise at 50–2000 Hz and the broadband noise above 6000 Hz are illustrated in Figure 9 and Figure 10. Obviously, the tonal noise of the loop-type propellers (#3 and #4) at all harmonics (the 2nd, the 4th, the 6th, the 8th, and the 10th shaft rate, etc.) is observed to be significantly reduced in comparison to the two-blade propeller (#1) and DJI Phantom III propeller (#5) under different rotational speeds. A pronounced peak of the sound pressure level (SPL) associated with the four-blade propeller tonal noise is observed at the harmonics of the 4th, the 8th, and the 12th shaft rate, etc., with the corresponding SPL showing a similar value to that of propeller #1. Furthermore, it is also seen that there exists a particularly low value of the specific harmonics of the 2nd, the 6th, and the 10th shaft rate, etc., likely caused by the symmetrical four-blade structure which resulted in suppressing the total tonal noise level [39]. Moreover, noticeable discrepancies are detected in the broadband noise in the high-frequency range among the four-blade propeller (#2), loop-type propellers (#3 and #4) and two-blade propeller (#1) (Figure 10). Propellers #2, #3, and #4 apparently present enhanced broadband noise components in the high-frequency range but low-level tonal noise components at the harmonics, while the propellers #1 and #5 produces high-level tonal noise components at the harmonics but low-level broadband noise components in the high-frequency range.

In addition, the tonal noise $OASPL_{i}$ at the ith shaft rate frequency ($f_{i},\;i=NB*1,\;2,\;3\ldots$) and the broadband noise $OASPL_{band}$ from $f_{L}$ to $f_{H}$ were calculated with Equations (6) and (7) to estimate the energy consumption of the tonal component and the average broadband noise. Obviously, as shown in Figure 11, the acoustics of the loop-type propellers (#3 and #4) and four-blade propeller (#2) are characterized by the lower-level tonal noise components at BPFs but the two-blade propellers (#1 and #5) by more tonal noise. The broadband noise in the high-frequency range of the four-blade propeller (#2) shows an obvious higher value compared to the loop-type propellers (#3 and #4) at all rotational speeds (Figure 12).

### 3.3. Objective Psychoacoustic Metrics

The noise loudness of the five propellers at the locations of Mics 1–4 was first compared in Figure 13a. With consideration that the 7th BPF harmonic (1260 Hz), a peak in the PSD spectrum, is primarily produced by the motor with seven pairs of magnets, we applied a notch filter of 1260 Hz in the data processing of the sound signal time series to remove the influence of motor noise. It is seen that the propellers #1 and #2 share the similar OASPL but the propeller #2 presents a pronounced improvement in noise loudness. This is probably because the four-blade propeller (#2) enables a reduction in the human-sensitive mid-frequency range noise while enhancing the human-unsensitive high-frequency range noise, as also observed at the propellers #3 and #4.

The same analysis was further undertaken for the five propellers at the stable thrust of $F_{z}=3\;N$. As shown in Figure 13b, while the loop-type propeller #3 had the highest rotational speed of 6148 rpm, it also presents a notably low loudness performance. The two-blade propellers (#1 and #5), however, show the highest loudness.

The noise sharpness was compared in Figure 14a: the four-blade propellers (#2, #3, and #4) present the maximum values at 5400 RPM, demonstrating a strong correlation with the average broadband noise $OASPL_{band}$ as shown in Figure 7, which is a result of the augmented high-frequency broadband noise due to the interaction of the multiple blades. The two-blade propellers (#1 and #5), however, show a low sharpness level, for instance, at the stable thrust of Fz=3 N (Figure 14b), displaying low values at high rotational speeds compared to the four-blade propellers (#2, #3 and #4). Furthermore, the increase in the high-frequency component shows a direction-independent feature.

The noise roughness, as shown in Figure 15, was used to estimate the noise strength at the BPF harmonics. The four-blade propellers (#2, #3 and #4) are observed to be capable of achieving a marked reduction in the roughness at $f_{mod}=180\;\mathrm{Hz}$ ($f_{mod}$ was required to be below 300 Hz in the roughness calculation), whereas the other two propellers show an increase. The roughness of the DJI Phantom propeller and two-blade propellers shows a higher value than that of the loop-type propeller and the wide chord loop-type propeller, because the loop-type structure reduces the BPF harmonics, hence leading to the lower roughness at $f_{mod}=180\;\mathrm{Hz}$. For the static thrust experiment, similar results for the roughness were found, featured by a trend of #5 > #1 > #4 > #3 > #2, but with no notable dependency on the rotation speed. In addition, the fluctuation strength (FS) of the five propellers is compared in Figure 16, which, however, shows very low values even for the fast rotational speeds of the five propellers, indicating that the FS may not play a crucial role in impacting the psychoacoustic annoyance.

The tonality of noise as shown in Figure 17 is used to evaluate the effect of frequency tone in terms of BPF harmonics, which is compared to the surrounding background sound. The loop-type propeller (3#) and wide chord loop-type propeller (4#) show the minimum values, consistently correlated with the tonal noise level ($OASPL_{i}$) (see Figure 11). It is worth noting that since only one single tonal peak is assessed per critical band in the calculation of tonality, the four-blade propeller model (#2) presents very high tonality with the minimum tonal peaks in the frequency domain. Additionally, the tonality shows direction-dependency with higher values at lower polar angles (Mic 1 and Mic 2), which is well consistent with the result based on the tonal noise emission theory by Made et al. (1970) [19].

The overall psychoacoustic performance (PA) is substantially illustrated in Figure 18 for the five propellers at different locations at 5400 RPM and the stable thrust of 3 N. The propeller #5 shows the highest PA value, indicating that this propeller poses the most annoying characteristics among the five propellers. Comparatively, the four-blade propellers (#2, #3 and #4) enable a notable improvement in PA because of the featured aeroacoustics of the multi-blade propeller as explained previously. Interestingly, even at the similar condition of OASPL, e.g., for Mics 1 and 2 at 5400 RPM, the loop-type propeller (#3) shows a significant improvement in PA. Furthermore, in Figure 19 we plotted the PA against thrust and figure of merit (FM) to compare the overall performance of the propellers. At a similar level of thrust, the loop-type propeller (#3) outperforms the other propellers in terms of PA but underperforms in FM of hover efficiency. This indicates that the loop-type propeller enables an improvement in the psychoacoustic performance but with paying a slight cost of lowering the aerodynamic efficiency.

## 4. Discussion

In this study, we proposed an integrated evaluation methodology of the aeroacoustics and psychoacoustics for a single propeller. Applying to five different typical propellers, we carried out an extensive investigation of the aerodynamic and acoustic characteristics and analyzed the psychoacoustic performance with the psychoacoustic annoyance (PA) models in terms of five psychoacoustic metrics of loudness, fluctuation strength, roughness, sharpness, and tonality. It is verified that the loop-type propeller enables a significant reduction in the tonal noise at BPF harmonics while increasing the broadband noise in high frequency, thus resulting in a remarkable improvement in PA according to Di’s model.

The suppression of the tonal noise at BPF harmonics have been reported in previous studies of axial flow fans [39,40], where a tonal noise reduction was verified due to the uneven circumferential spacing of blades, and the periodic fluctuation was recognized as the major source of tonal components. Here the distinctive acoustic characteristics are inferred to be likely associated with the featured reduced structural symmetry of the loop-type propeller, which contributes to the thrust generation though the four blades (two front blades and two rear blades), of which the tips of two adjacent blades are connected forming two circle loops, rather than by two completely symmetrical blades. Thus, the interval angle between front and rear blade is responsible for the improvement of the acoustic performance, leading to the pronounced peaks in a cycle, which is distinguished from those of the blade passing frequency harmonics. This finding is supported by the experimental study by Cattanei (2021) [41] in association with the aeroacoustics of a fan uneven blade.

The loop-type propellers are further verified to be capable of enhancing the broadband noise in high frequencies, which may be due to the turbulence flows particularly induced by the loop-type propellers (#3 and #4) and the four-blade propeller (#2) in comparison to the two-blade propellers (#1 and #5). Wu (2021) [42] also reported that the broadband sound power level of a stator can increase with an increase in the number of blades. Interestingly, it is found that the broadband noise exceeding 6000 Hz of the four-blade propeller (#2) is noticeably higher than that of the loop-type propellers (#3 and #4) (Figure 12). This may be reasonably explained by the featured morphology of the loop-type propellers, which suppress the wingtip vortex, and hence the vortex-dominated flow interaction, in comparison to the four individual blades, whereas the broadband noise suppression cannot be achieved by the conventional uneven multiple blade structures, as reported in previous studies of the axial flow fans [39,40,41].

With respect to the psychoacoustic performance, we performed a comprehensive analysis of the improved Zwicker’s psychoacoustic metrics. While several previous studies [24,26,43] pointed out that the human response to drone noise highly depends on the dominant factor of loudness metrics, in this study, it is verified that the loop-type propellers can markedly reduce the mid-range-frequency tonal noise at the cost of broadband noise in high-frequency range, which is beneficial for the loudness of low weighting in the high-frequency range. Gwak et al. [24] also reported that the fluctuation strength and the sharpness are a notable feature observed in the annoyance of UAV noise. While the higher broadband noise is observed at high frequency in the loop-type propeller with a noticeable increase in sharpness, an apparent reduction is observed in the prominence of the tonal peaks (i>25∗NB) (Figure 7). This result was supported by an experimental study by Cattanei et al. [41]. In addition, it is worth noting that the low-frequency modulation noise (<4 Hz), i.e., the source of the fluctuation strength, is normally caused by a slightly different rotational speed (<4 RPS) of the multiple propellers rather than the isolated propellers.

Regarding the limitations in the current study, it is worth noting that the acoustic characteristics evaluated here in the in-house hovering conditions could change in real field flights of multirotor drones. Firstly, the sound attenuation rate of the high-frequency component turns out to be higher in air compared to the low-frequency component, which may reduce the negative effect on the broadband noise (>6 kHz) increase in loop-type propellers [44]. Secondly, the tonal noise suppression at mid-frequency could be unaffected. This phenomenon may be capable of enhancing the improvement of the aeroacoustic characteristics in OASPL and the loudness of psychoacoustic performance. Thirdly, the modulation effect associated with the tonal noise induced by different propellers of the same multirotor drone likely occurs, which may alter the psychoacoustic metrics of the roughness over a range of 15–300 Hz and a fluctuation strength less than 20 Hz. Moreover, while the loop-type propeller enables suppression of the tonal noise in terms of the annoyance reduction in roughness and fluctuations strength, it has more mass/weight than other propellers, which may result in additional aerodynamic drag cost (or moment of inertia) for rotational speed control in real field flights. Future experiments and analyses will be undertaken to explore these issues by using field flight tests of multirotor drones.

## 5. Conclusions

In this paper, we proposed an integrated evaluation methodology and carried out a comprehensive analysis of the aerodynamic, aeroacoustic and psychoacoustic characteristics of a two-blade propeller, four-blade propeller, loop-type propeller, wide chord loop-type propeller, and DJI Phantom III propeller. A force–torque cell and microphones were used to measure and compare the aerodynamic and aeroacoustic performance of the five propellers, and an objective psychoacoustic evaluation was employed to conduct psychologic evaluation of the aerodynamic noise in an anechoic chamber. Our main findings are summarized as follows:The loop-type propeller shows a comparatively higher aeroacoustic performance, capable of achieving a remarkable reduction of a maximum of 4.7 dB in the overall sound pressure level (OASPL) at 5400 RPM and a maximum of 3.5 dB in the OASPL at $F_{z}=3\;N$ compared to the DJI Phantom III propeller. Our results demonstrate that the loop-type propeller, as an innovative drone propeller design, presents notable characteristics in the tonal noise components for harmonics suppression, particularly at the high-order (i>25∗NB) harmonics in the high-frequency range.It is verified that the loop-type structure enables the suppression of the tip vortex and the near-field turbulence while its broadband noise sound pressure level (SPL) shows notably higher values compared to the two-blade propeller.Utilizing the improved Zwicker’s psychoacoustic annoyance (PA) model by Di et al. (2016) [31] we find that the loop-type propeller enables lowering loudness, tonality, and roughness, hence leading to a pronounced improvement in the psychoacoustic annoyance (PA), even under similar condition of OASPL.Through directly measuring both thrusts and torques, it is verified that the loop-type propeller exhibits a slightly impaired aerodynamic performance in terms of the figure of merit (FM) and thrust coefficient ($C_{t}$), thus resulting in an increase in the power and rotational speed at some specific thrusts.

Our results thus indicate that the loop-type propeller can provide a prototype low-noise design for drone propellers in terms of noise OASPL reduction and psychoacoustic annoyance. This study points to the usefulness and validity of the integrated evaluation methodology of aeroacoustics and psychoacoustics in the design of low-noise propellers in association with multirotor drones.

## Figures and Tables

**Figure 1 ijerph-20-01955-f001:**
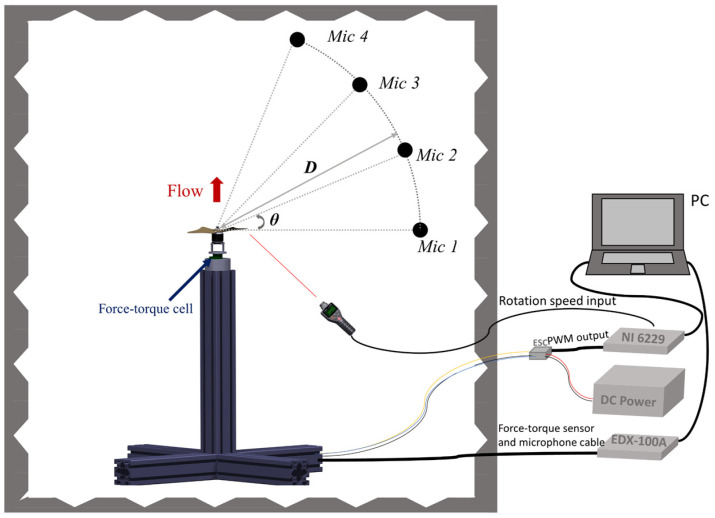
The hover stand setup in an anechoic chamber for aerodynamic and acoustic measurements.

**Figure 2 ijerph-20-01955-f002:**
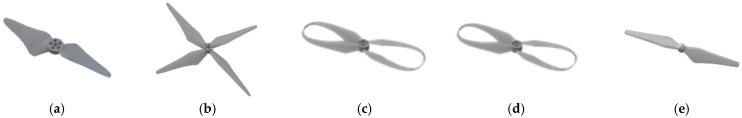
Five propellers: (**a**) two-blade propeller (#1), (**b**) four-blade propeller (#2), (**c**) loop-type propeller (#3), (**d**) wide chord loop-type propeller (#4), and (**e**) DJI Phantom propeller (#5).

**Figure 3 ijerph-20-01955-f003:**
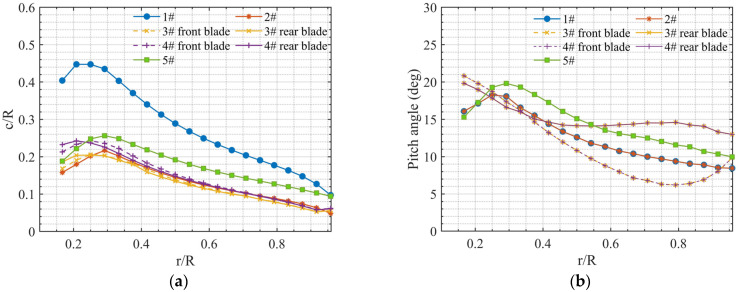
Radial distributions of chord length (**a**) and pitch angle (**b**) of five propellers.

**Figure 4 ijerph-20-01955-f004:**
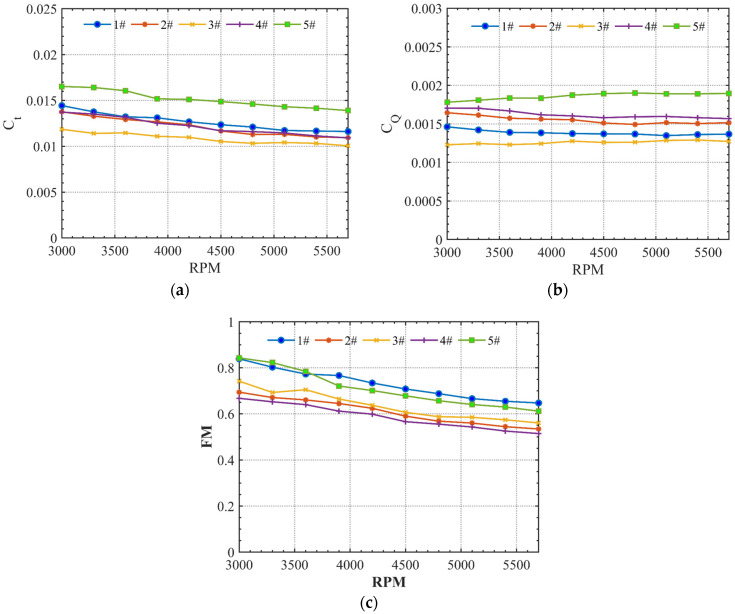
Comparison of (**a**) the thrust coefficient ($C_{t}$), (**b**) torque coefficient ($C_{Q}$), and (**c**) figure of merit (FM) among the two-blade propeller (#1), four-blade propeller (#2), loop-type propeller (#3), wide chord loop-type propeller (#4), and DJI Phantom III propeller (#5).

**Figure 5 ijerph-20-01955-f005:**
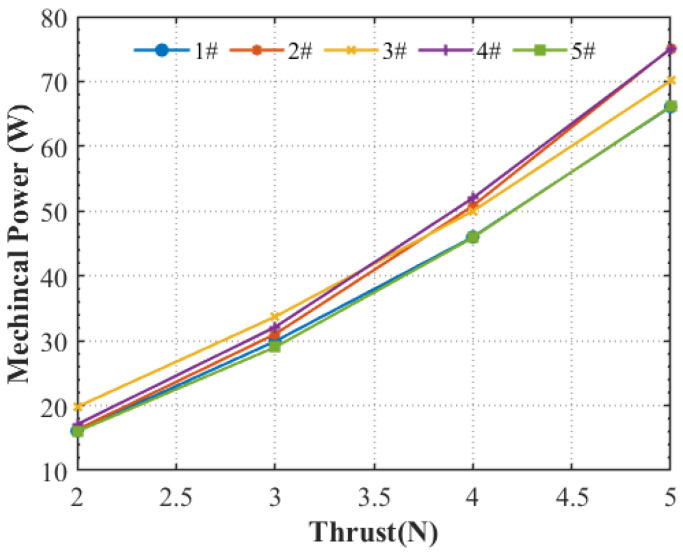
Comparison of mechanical power vs. thrust at four stable thrusts of five propellers.

**Figure 6 ijerph-20-01955-f006:**
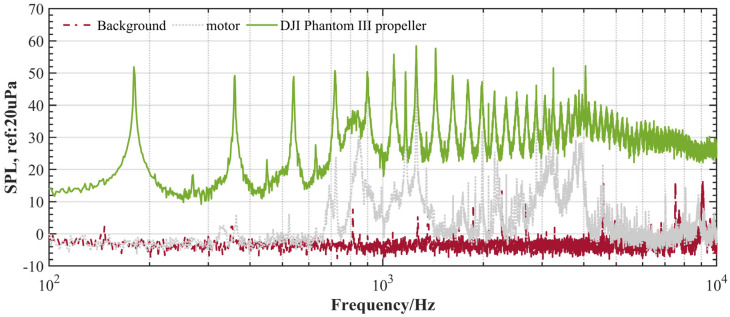
Comparison of the power spectral density of background noise, unloaded motor noise, and DJI Phantom III propeller (#5) (5400 RPM, θ=25 deg).

**Figure 7 ijerph-20-01955-f007:**
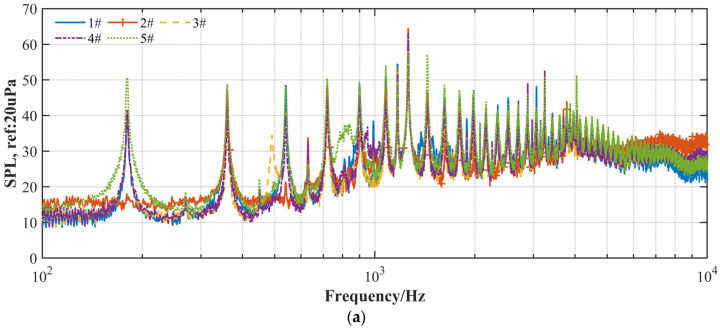
Power spectral density of five propellers at a rotational speed of 5400 RPM at Mic 2. (**a**) 100–20 kHz, (**b**) 100–4 kHz, and (**c**) 4.5–10 kHz.

**Figure 8 ijerph-20-01955-f008:**
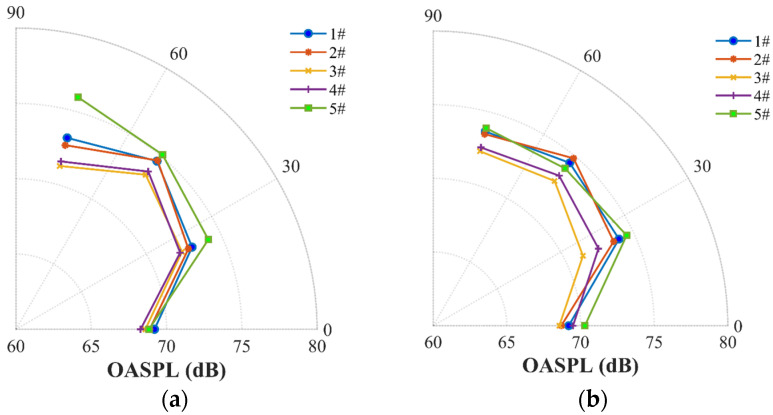
Comparison of OASPL among the two-blade propeller (#1), four-blade propeller (#2), loop-type propeller (#3), wide chord loop-type propeller (#4), and DJI Phantom III propeller (#5) at a rotational speed of 5400 rpm (**a**) and at a stable thrust of 3 N (**b**).

**Figure 9 ijerph-20-01955-f009:**
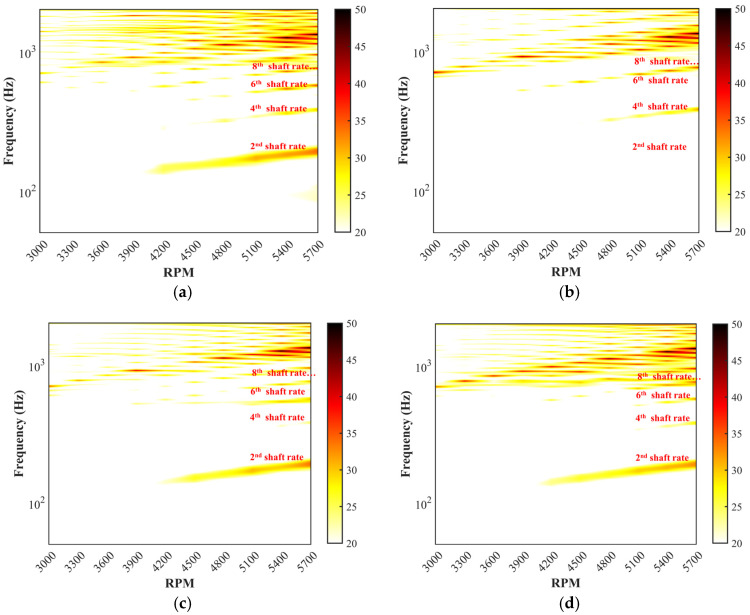
Contour maps of the SPL in dB (Mic 2, θ=25 deg) over a frequency range of 50–2000 Hz associated with propeller noises: (**a**) two-blade propeller (#1), (**b**) four-blade propeller (#2), (**c**) loop-type propeller (#3), (**d**) wide chord loop-type propeller (#4), and (**e**) DJI Phantom propeller (#5) at various rotational speeds.

**Figure 10 ijerph-20-01955-f010:**
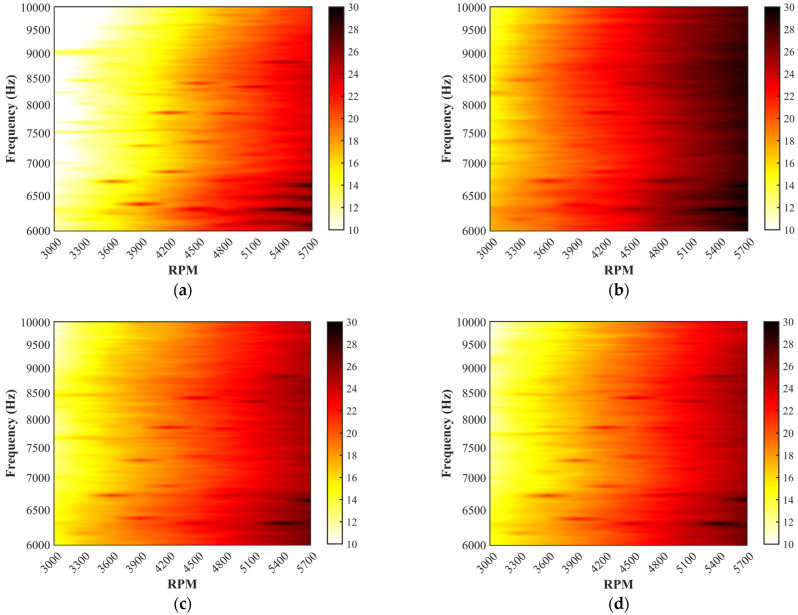
Contour maps of the SPL in dB (Mic 2, θ=25 deg) over a frequency range of 6000–10,000 Hz associated with propeller noises: (**a**) two-blade propeller (#1), (**b**) four-blade propeller (#2), (**c**) loop-type propeller (#3), (**d**) wide chord loop-type propeller (#4), and (**e**) DJI Phantom propeller (#5) at various rotational speeds.

**Figure 11 ijerph-20-01955-f011:**
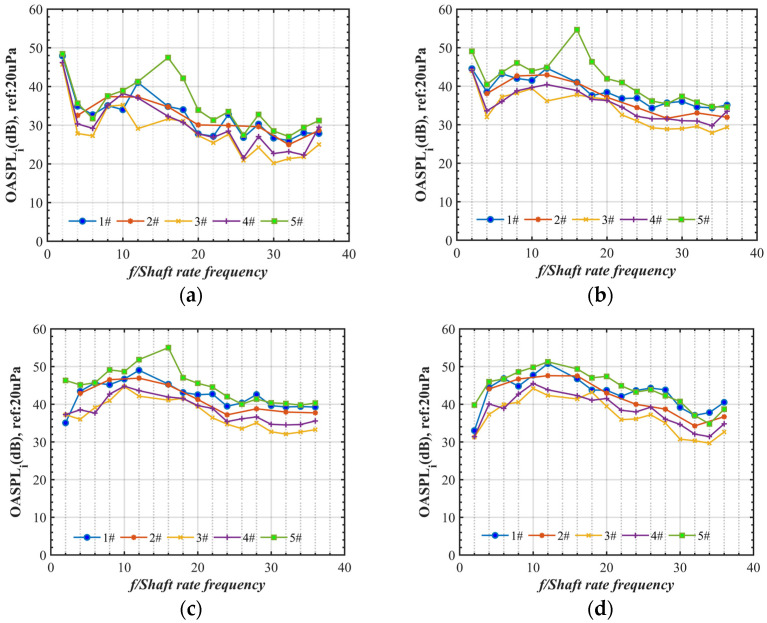
Comparison of the tonal noise, $OASPL_{i}\;\left(i=2,\;4,\ldots ,\;36\right)$, below 5400 rpm at four locations of Mic 1 (**a**), Mic 2 (**b**), Mic 3 (**c**), and Mic 4 (**d**).

**Figure 12 ijerph-20-01955-f012:**
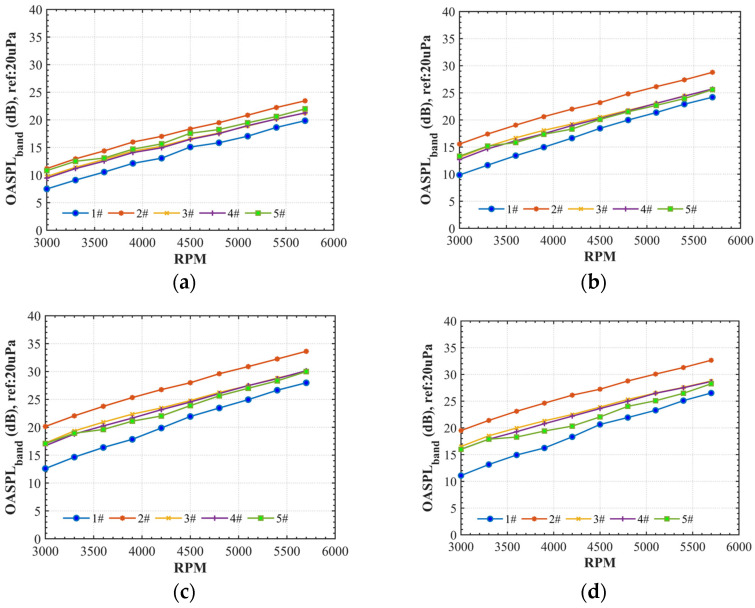
Comparison of the average broadband noise ($OASPL_{band}$) beyond 6000 Hz vs. rotation speed at four locations of Mic 1 (**a**), Mic 2 (**b**), Mic 3 (**c**), and Mic 4 (**d**).

**Figure 13 ijerph-20-01955-f013:**
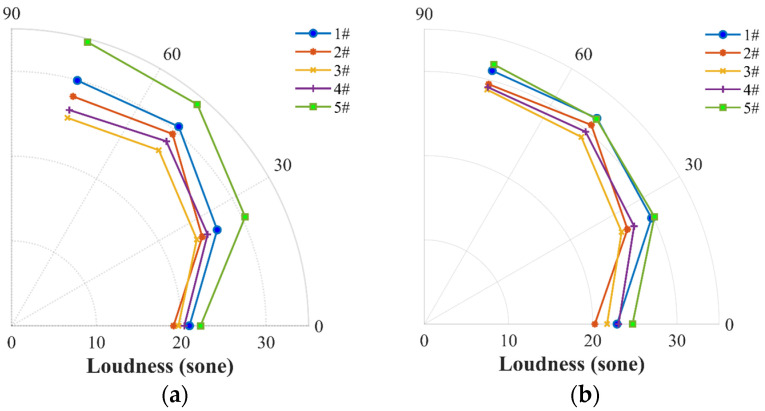
Comparison of noise loudness among the two-blade propeller (#1), four-blade propeller (#2), loop-type propeller (#3), wide chord loop-type propeller (#4), and DJI Phantom III propeller (#5) at a rotational speed of 5400 rpm (**a**) and at a stable thrust of 3 N (**b**).

**Figure 14 ijerph-20-01955-f014:**
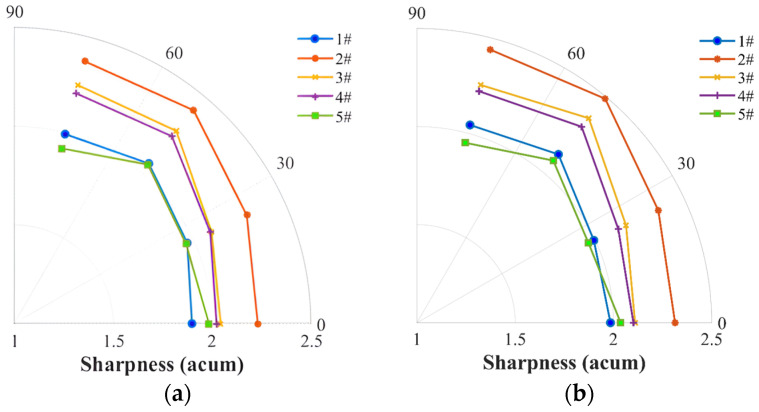
Comparison of noise sharpness among the two-blade propeller (#1), four-blade propeller (#2), loop-type propeller (#3), wide chord loop-type propeller (#4), and DJI Phantom III propeller (#5) at a rotational speed of 5400 rpm (**a**) and at a stable thrust of 3 N (**b**).

**Figure 15 ijerph-20-01955-f015:**
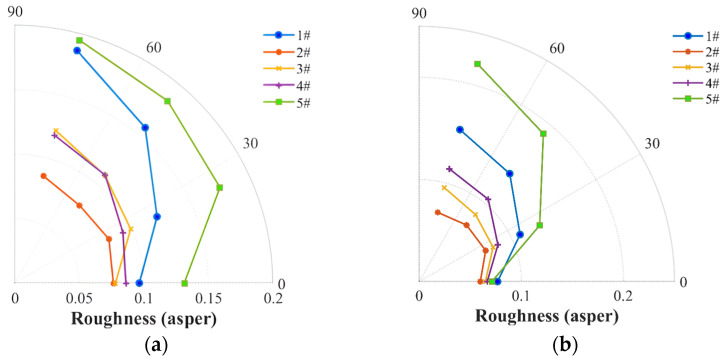
Comparison of noise roughness among the two-blade propeller (#1), four-blade propeller (#2), loop-type propeller (#3), wide chord loop-type propeller (#4), and DJI Phantom III propeller (#5) at a rotational speed of 5400 rpm (**a**) and at a stable thrust of 3 N (**b**).

**Figure 16 ijerph-20-01955-f016:**
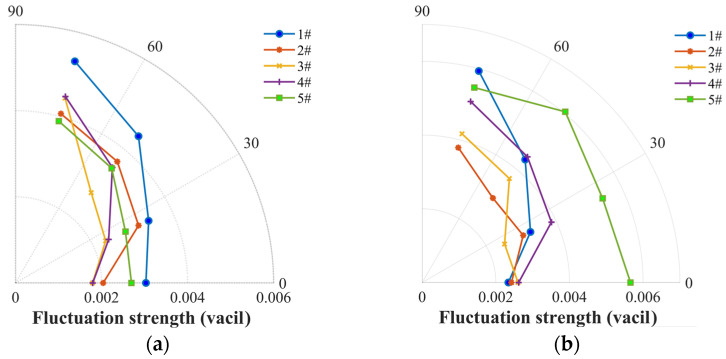
Comparison of fluctuation strength among the two-blade propeller (#1), four-blade propeller (#2), loop-type propeller (#3), wide chord loop-type propeller (#4), and DJI Phantom III propeller (#5) at a rotational speed of 5400 rpm (**a**) and at a stable thrust of 3 N (**b**).

**Figure 17 ijerph-20-01955-f017:**
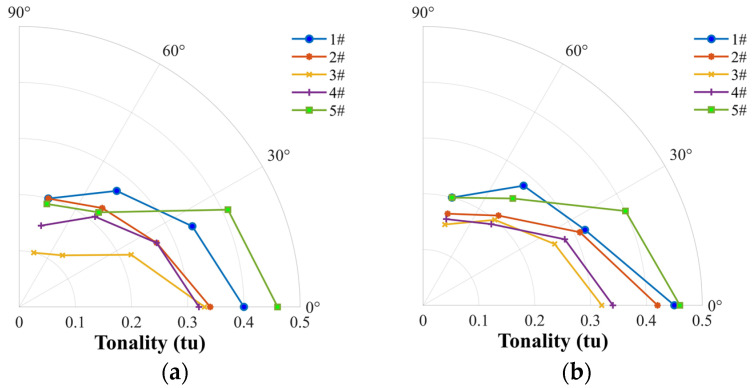
Comparison of tonality among the two-blade propeller (#1), four-blade propeller (#2), loop-type propeller (#3), wide chord loop-type propeller (#4), and DJI Phantom III propeller (#5) at a rotational speed of 5400 rpm (**a**) and at a stable thrust of 3 N (**b**).

**Figure 18 ijerph-20-01955-f018:**
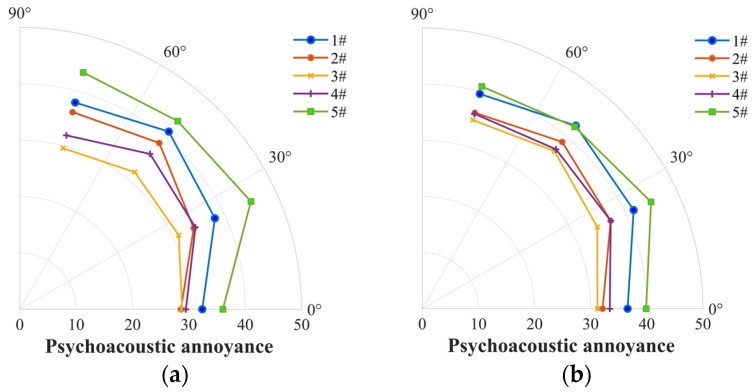
Comparison of psychoacoustic annoyance (Di’s model) among the two-blade propeller (#1), four-blade propeller (#2), loop-type propeller (#3), wide chord loop-type propeller (#4), and DJI Phantom III propeller (#5) at a rotational speed of 5400 rpm (**a**) and at a stable thrust of 3 N (**b**).

**Figure 19 ijerph-20-01955-f019:**
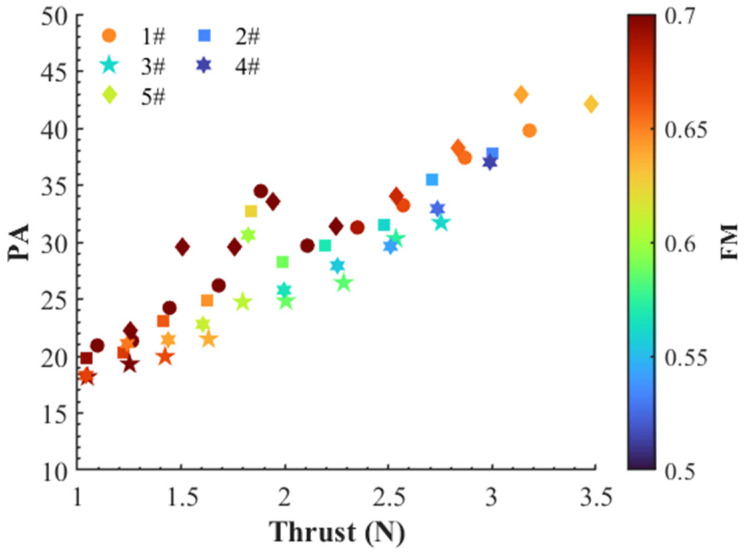
Mean psychoacoustic annoyance (Di’s model) vs. thrust and FM among the two-blade propeller (#1), four-blade propeller (#2), loop-type propeller (#3), wide chord loop-type propeller (#4), and DJI Phantom III propeller (#5) at a rotational speed of 3000–5700 rpm.

## Data Availability

The data will be available on request to the corresponding author’s email with appropriate justification.

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
