# Peer review of "Integrated Evaluation of the Aeroacoustics and Psychoacoustics of a Single Propeller"

_ijerph, 2023, doi:10.3390/ijerph20031955_

Round 1

Reviewer 1 Report

Dear authors, thank you for sending this manuscript. The contents are well explained, however, little observations can be made. Please find the following:

1. Figure 5 if printed in black and white is not clearly readable. Please find another way of how to show data.

2. Figure 6. same comments as per above. please use dashed or dotted style for one line and leave the other one continuous

3. figure 7. as per above comments

4. please review the english of the all text 

Thank you

Reviewer 2 Report

Dear author,

I think it is a good work and deserves to be published in the journal, but I think that, beforehand, it is necessary to clarify some questions:

1) FM is defined in equation (1). Figure or factor of merit?

2) In equation (1) and (2), A' is? (On line 117 A is defined, not A').

3) The definition of sound pressure level and power spectral density in equations (4) and (5) is not the usual one in the literature. Can you clarify this issue?

Thank you very much

Reviewer 3 Report

The authors present an interesting aeroacoustic and aerodynamic benchmark of five different propellers for UAV applications. A relevant added feature in the acoustic analysis is a psychoacoustic annoyance estimation based on existing models. The manuscript is well written and well structured, and only some minor comments need to be accounted for before I recommend publication. Henceforth, PXXLYY refers to page XX and line YY.

General comments:

I am quite surprised that the authors did not consider the tonality metric (from models like Aures or Terhardt) in propeller applications since tonality is one of the main features of their spectra (see Fig 7 for example). Therefore, I strongly recommend extending the psychoacoustic analysis to also include the tonality, which can be included in the global psychoacoustic annoyance metric via newer models like those by More and Di et al:

[1] More, S.R., Aircraft Noise Characteristics and Metrics. Purdue University; 2010. Report No. PARTNER–COE–2011–004.

[2] Di, G.Q., et al., Improvement of Zwicker’s psychoacoustic annoyance model aiming at tonal noises. Applied Acoustics. 2016;105:164-70.

What were the mass and estimated drag of the propellers? I can imagine that the loop-type propellers are heavier and have additional drag compared to the conventional designs. Please include this in your analysis to have a fair comparison.

Ideally, some comments/analysis on what is the expected aeroacoustic, aerodynamic and psychoacoustic performance of these three propellers in realistic operational conditions (i.e. not just hover) would be included in the manuscript for higher relevance.

I strongly recommend adding a graph (or set of graphs) showing the psychoacoustic annoyance estimated on the Y-axis and the aerodynamic performance on the X-axis (either the thrust, the FM, or another metric). This way, the reader can clearly compare the overall performance of the propellers.

P02L85 - I would recommend referring to a similar study that considered the psychoacoustic performance of different designs of wind turbine blades for community noise impact as a similar approach in a different application:

[3] Merino-Martinez, R, et al., Holistic approach to wind turbine noise: From blade trailing-edge modifications to annoyance estimation. Renewable and Sustainable Energy Reviews. 2021;148:1-14.

P03L102 - Please specify the cutoff frequency and background noise level (ideally a spectrum in a figure) of the anechoic chamber employed.

Figure 1: Why did you not study the downward (theta < 0) polar angles? These are the most relevant for community noise for flying propellers. This is one of the main pitfalls of this manuscript. 

Figure 7c: Why are shaded areas (I assume that is the standard deviation of the spectra) only on this figure?

Figure 16: The unit in the X axis should read vacil and not vacils.

Figure 17b: There is a typo in the X axis label.

Author Response

Reply to reviewer #3

We would like to acknowledge the invaluable comments and insightful suggestions of the reviewer. We have revised our manuscript and addressed all the reviewer comments in an itemized reply with the changes highlighted in the revised manuscript. We sincerely hope that the revisions would be able to address all the concerns raised by the reviewer.

  1. I am quite surprised that the authors did not consider the tonality metric (from models like Aures or Terhardt) in propeller applications since tonality is one of the main features of their spectra (see Fig 7 for example). Therefore, I strongly recommend extending the psychoacoustic analysis to also include the tonality, which can be included in the global psychoacoustic annoyance metric via newer models like those by More and Di et al:

[1] More, S.R., Aircraft Noise Characteristics and Metrics. Purdue University; 2010. Report No. PARTNER–COE–2011–004.

[2] Di, G.Q., et al., Improvement of Zwicker’s psychoacoustic annoyance model aiming at tonal noises. Applied Acoustics. 2016;105:164-70.

We thank this reviewer for pointing out this important issue. Accordingly, we revised the relevant text in the sections of Method (on page 7, line 249) and Results (on page 18, line 468, line 485). The results of tonality metric (Aures) were added as shown in Figure 17, which are well consistent with the trend of and the tonal noise emission characteristics [19]. We further made some revisions on the illustration of psychoacoustic annoyance in Figure 18 and 19, based on Di’s model by taking account of the tonality effect. Considering that only one single tonal component (harmonic) per critical band is assessed in the calculation of Aures tonality, we keep using the to evaluate the variance of the tonal noise component among the five propellers in section 3.2.2.

  1. What were the mass and estimated drag of the propellers? I can imagine that the loop-type propellers are heavier and have additional drag compared to the conventional designs. Please include this in your analysis to have a fair comparison.

In principle, the mass (including the moment of inertia) of the propellers does not exert substantial influence on the drag in the current hover condition, when the rational speed is set to be stationary, i.e., , according to the equation of rotational motion, such as:

where  is the torque of motor, identical to the overall drag of the propellers,  is the equal torque induced by the aerodynamic drag,  is the moment of inertia, and  is the rotation speed in rad/s. However, the heavier loop-type propellers do increase the overall mass / weight of the multi-rotor drone, which may bring some additional cost in the rotational speed control, resulting in  in real field flights. We have accordingly added some relevant discussion on this issue on page 20, line 556.

  1. Ideally, some comments/analysis on what is the expected aeroacoustic, aerodynamic and psychoacoustic performance of these three propellers in realistic operational conditions (i.e. not just hover) would be included in the manuscript for higher relevance.

We revised the relevant discussion accordingly on page 20, line 544.

  1. I strongly recommend adding a graph (or set of graphs) showing the psychoacoustic annoyance estimated on the Y-axis and the aerodynamic performance on the X-axis (either the thrust, the FM, or another metric). This way, the reader can clearly compare the overall performance of the propellers.

Very good suggestion. We added Figure 19 to illustrate this and revised the relevant text on page18, line 485. In Figure 19, we made a clarification of the overall performance of each propeller to show that the loop-type propeller enables an improvement in the psychoacoustic performance but with a cost in lowering the aerodynamic efficiency.

  1. P02L85 - I would recommend referring to a similar study that considered the psychoacoustic performance of different designs of wind turbine blades for community noise impact as a similar approach in a different application:

[3] Merino-Martinez, R, et al., Holistic approach to wind turbine noise: From blade trailing-edge modifications to annoyance estimation. Renewable and Sustainable Energy Reviews. 2021;148:1-14.

This is fixed accordingly.

  1. P03L102 - Please specify the cutoff frequency and background noise level (ideally a spectrum in a figure) of the anechoic chamber employed.

We revised the relevant text accordingly on page 3, line 104 and added the background noise spectrum in Figure 6.

  1. Figure 1: Why did you not study the downward (theta < 0) polar angles? These are the most relevant for community noise for flying propellers. This is one of the main pitfalls of this manuscript.

The microphone array was designed and located downstream of the propeller as shown of the induced slipstream in Figure 1, which are considerably relevant for the community noise for flying propellers. To clarify this issue, we revised the relevant description of this issue on page 2, line 112.

  1. Figure 7c: Why are shaded areas (I assume that is the standard deviation of the spectra) only on this figure?

The shaded areas (standard deviation) are used to enhance the readability and to highlight the discrepancy among the five propellers in terms of the spectrum of broadband noise (>4.5kHz) where the PSD shows large fluctuations on the logarithmic scale.

  1. Figure 16: The unit in the X axis should read vacil and not vacils.

This is fixed accordingly.

  1. Figure 17b: There is a typo in the X axis label

This is fixed accordingly.
